# Safe Reinforcement Learning
# with Natural Language Constraints

**Tsung-Yen Yang**[*]
Princeton University
ty3@princeton.edu

**Michael Hu**[*]
Princeton University
michael.hu@yobi.ventures

**Yinlam Chow**
Google Research
yinlamchow@google.com

**Peter J. Ramadge**
Princeton University
ramadge@princeton.edu

**Karthik Narasimhan**
Princeton University
karthikn@princeton.edu

## Abstract

While safe reinforcement learning (RL) holds great promise for many practical applications like robotics or autonomous cars, current approaches require specifying constraints in mathematical form. Such specifications demand domain expertise, limiting the adoption of safe RL. In this paper, we propose learning to interpret natural language constraints for safe RL. To this end, we first introduce HAZARD-WORLD, a new multi-task benchmark that requires an agent to optimize reward while not violating constraints specified in free-form text. We then develop an agent with a modular architecture that can interpret and adhere to such textual constraints while learning new tasks. Our model consists of **(1)** a *constraint interpreter* that encodes textual constraints into spatial and temporal representations of forbidden states, and **(2)** a *policy network* that uses these representations to produce a policy achieving minimal constraint violations during training. Across different domains in HAZARDWORLD, we show that our method achieves higher rewards (up to 11x) and fewer constraint violations (by 1.8x) compared to existing approaches. However, in terms of absolute performance, HAZARDWORLD still poses significant challenges for agents to learn efficiently, motivating the need for future work.[1]

## 1   Introduction

Although reinforcement learning (RL) has shown promise in several simulated domains such as games [1, 2, 3] and autonomous navigation [4, 5], deploying RL in real-world scenarios remains challenging [6]. In particular, real-world RL requires ensuring the safety of the agent and its surroundings, which means accounting for *constraints* during training that are orthogonal to maximizing rewards. For example, a cleaning robot must be careful to not knock the television over, even if the television lies on the optimal path to cleaning the house.

Safe RL tackles these challenges with algorithms that maximize rewards while simultaneously minimizing constraint violations during exploration [7, 8, 9, 10, 11, 12, 13, 14, 15, 16]. However, these algorithms have two key limitations that prevent their widespread use. First, they require us

---

[*] Equal contribution.
[1] Code and data are available at https://github.com/princeton-nlp/SRL-NLC

35th Conference on Neural Information Processing Systems (NeurIPS 2021).

to provide constraints in mathematical or logical forms, which calls for specific domain expertise. Second, a policy trained with a specific set of constraints cannot be transferred easily to learn new tasks with the same set of constraints, since current approaches do not maintain an explicit notion of constraints separate from reward-maximizing policies. This means one would have to retrain the policy (with constraints) from scratch.

We consider the use of *natural language* to specify constraints (which are orthogonal to rewards) on learning. Human languages provide an intuitive and easily-accessible medium for describing constraints–not just for machine learning experts or system developers, but also for potential end users interacting with agents such as household robots. Consider the environment in Fig. 1 for example. Instead of expressing a constraint as $\sum_{t=0}^{T} \mathbf{1}_{s_t \in \text{lava}} \cdot \mathbf{1}_{\nexists s_{t'} \in \text{water}, \; t' \in [0,1,...,t-1]} = 0$, one could simply say "*Do not visit the lava before visiting the water*". The challenge of course, lies in training the RL agent to accurately interpret and adhere to the textual constraints as it learns a policy for the task.

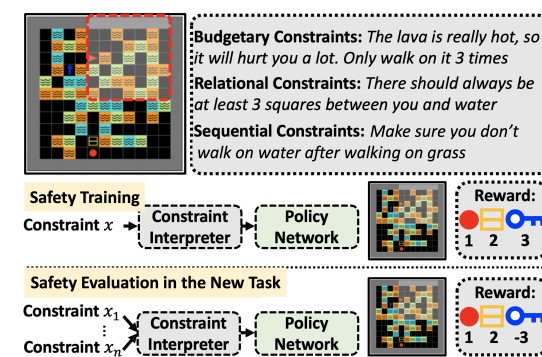

Figure 1: Learning to navigate with language constraints. The figure shows **(1)** a third-person view of the environment (red dotted square box), **(2)** three types of language constraints, **(3)** items which provide rewards when collected. During safety training, the agent learns to interpret textual constraints while learning the task (*i.e.,* collect rewards). During safety evaluation, the agent learns a new task with different rewards while following the constraints and minimizing violations.

To study this problem, we first create HAZARDWORLD, a collection of grid-world and robotics environments for safe RL with textual constraints (Fig. 1). HAZARDWORLD consists of separate '*safety training*' and '*safety evaluation*' sets, with disjoint sets of reward functions and textual constraints between training and evaluation. To do well on HAZARDWORLD, an agent has to learn to interpret textual constraints during safety training and safely adhere to any provided constraints while picking up new tasks during the safety evaluation phase. Built on existing RL software frameworks [17, 18], HAZARDWORLD consists of navigation and object collection tasks with diverse, crowdsourced, free-form text specifying three kinds of constraints: **(1)** *budgetary* constraints that limit the frequency of being in unsafe states, **(2)** *relational* constraints that specify unsafe states in relation to surrounding entities, and **(3)** *sequential* constraints that activate certain states to be unsafe based on past events (*e.g.,* "*Make sure you don't walk on water after walking on grass*"). Our setup differs from instruction following [19, 20, 21, 22, 23, 24] in two ways. First, instructions specify what to do, while textual constraints only inform the agent on what *not to do*, independent of maximizing rewards. Second, learning textual constraints is a means for ensuring safe exploration while adapting to a new reward function.

In order to demonstrate learning under this setting, we develop ***P**olicy **O**ptimization with **L**anguage **CO**nstraints* (POLCO), where we disentangle the representation learning for textual constraints from policy learning. Our model first uses a *constraint interpreter* to encode language constraints into representations of forbidden states. Next, a *policy network* operates on these representations and state observations to produce actions. Factorizing the model in this manner allows the agent to retain its constraint comprehension capabilities while modifying its policy network to learn new tasks.

Experiments demonstrate that our approach achieves higher rewards (up to 11x) while maintaining lower constraint violations (up to 1.8x) compared to several baselines on two different domains within HAZARDWORLD. Nevertheless, HAZARDWORLD remains far from being solved, especially in tasks with high-dimensional observations, complex textual constraints and those requiring high-level planning or memory-based systems.

## 2   Related Work

**Safe RL.** Safe RL deals with learning constraint-satisfying policies [25], or learning to maximize rewards while minimizing constraint violations.[2] This is a constrained optimization problem, and thus different from simply assigning negative reward values to unsafe states. Furthermore, large negative reward values for constraint violations can destabilize training and lead to degenerate behavior, such as the agent refusing to move. In prior work, the agent typically learns policies either by **(1)** exploring the environment to identify forbidden behaviors [7, 27, 8, 9, 28], or **(2)** using expert demonstration data to recognize safe trajectories [29, 30, 31, 10]. All these works require a human to specify the cost constraints in mathematical or logical form, and the learned constraints cannot be easily reused for new learning tasks. In this work, we design a modular architecture to learn to interpret textual constraints, and demonstrate transfer to new learning tasks.

**Instruction following.** Our work closely relates to the paradigm of instruction following in RL, which has previously been explored in several environments [19, 32, 20, 33, 21, 34, 35, 36, 37, 38, 39]. Prior work has also focused on creating realistic vision-language navigation datasets [40, 41, 4, 42] and proposed computational models to learn multi-modal representations that fuse images with goal instructions [43, 44, 45, 46, 47, 48, 49, 50, 51]. Our work differs from the traditional instruction following setup in two ways: **(1)** Instruction following seeks to (roughly) 'translate' an instruction directly into an action policy. This does not apply to our setting since the textual constraints only tell an agent what *not to do*. To actually obtain rewards, the agent has to explore and figure out optimal policies on its own. **(2)** Since constraints are decoupled from rewards and policies, agents trained to understand certain constraints can transfer their understanding to respect these constraints in new tasks, even when the new optimal policy is drastically different. Therefore, we view this work as orthogonal to traditional instruction following–one could of course combine both instructions and textual constraints to simultaneously advise an agent on what to do and what not to do.

**Connection to Seldonian algorithms.** POLCO can also be interpreted as a Seldonian algorithm [52]. Seldonian algorithms ensure ML safety through three steps: (1) defining a goal, (2) defining an interface for users to provide constraints, and (3) creating an algorithm that satisfies the goal and constraints. Here, we use natural language as the interface for end users and map natural language into optimization constants and vector representations. Thus, POLCO is also a potential step towards widely developing and deploying Seldonian algorithms.

**Constraints in natural language.** Our notion of 'constraints' in this paper differs from prior work that uses instructions to induce planning constraints [33, 53, 39]–these works again provide instructions for the agent on how to perform the task. Perhaps closest to this paper is the work of Misra et al. [5], which proposes datasets to study spatial and temporal reasoning, containing a subset focusing on *trajectory constraints* (*e.g.,* "*go past the house by the right side of the apple*"). However, they do not disentangle the rewards from the constraints, which may be orthogonal to each other. Prakash et al. [54] train a constraint checker to identify whether a constraint (specified in text) has been violated in a trajectory. While their motivation is similar, they ultimately convert constraints to negative rewards, whereas we use a modular approach that allows disentangling reward maximization from minimizing constraint violations and is compatible with modern algorithms for safe RL.

## 3   Preliminaries

**Problem formulation.** Our learning problem can be viewed as a partially observable constrained Markov decision process [55], which is defined by the tuple $< \mathcal{S}, \mathcal{O}, \mathcal{A}, T, Z, \mathcal{X}, R, C >$. Here $\mathcal{S}$ is the set of states, $\mathcal{O}$ is the set of observations, $\mathcal{A}$ is the set of actions, $T$ is the conditional probability $T(s'|s, a)$ of the next state $s'$ given the current state $s$ and the action $a$, and $Z$ is the conditional probability $Z(o|s)$ of the observation $o$ given the state $s$. In addition, $\mathcal{X}$ is the set of textual constraint specifications, $R : \mathcal{S} \times \mathcal{A} \to \mathbb{R}$ is the reward function, which encodes the immediate reward provided when the agent takes an action $a$ in state $s$, and $C : \mathcal{S} \times \mathcal{A} \times \mathcal{X} \to \mathbb{R}$ is the true underlying constraint function described by $x \in \mathcal{X}$, which specifies positive penalties for constraint violations due to an action $a$ in a state $s$. Finally, we assume each $x \in \mathcal{X}$ corresponds to a specific cost function $C$.

---

[2]In this paper, we consider minimizing constraint violations in expectation [9] and leave stricter notions such as enforcing zero violations [26] to future work.

| Constraint | Examples | |
|---|---|---|
| Budgetary | *Lava hurts a lot, but you have special shoes that you can use to walk on it, but only up to 5 times, remember!* | 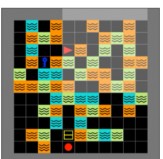 |
| Relational | *Water will hurt you if you are two steps or less from them.* | |
| Sequential | *Make sure you don't walk on water after walking on grass.* | |

(a) HAZARDWORLD-grid

| Constraint | Examples | |
|---|---|---|
| Budgetary | *Do not enter the blue square. It is safe to never cross at all.* | 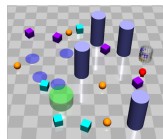 |
| Relational | *Three feet is the minimum distance to all the dark blue circles.* | |
| Sequential | *Once a purple box gets touched dark blue circles are disallowed.* | |

(b) HAZARDWORLD-robot

Table 1: Examples of textual constraints for HAZARDWORLD-grid and HAZARDWORLD-robot. **(a)** An agent (red triangle) seeks to collect the reward entity (ball, box, key) while avoiding the cost entity (lava, water, grass). **(b)** An agent (red point) aims to reach a goal position (green area) while avoiding the obstacles (vases, pillars, cubes, *etc.*). Please see the supplementary material for more details.

**RL with constraints.** The goal of the learning agent is to acquire a good control policy that maximizes rewards, while adhering to the specified constraints as much as possible during the learning process. Thus, the agent learns a policy $\pi : \mathcal{O} \times \mathcal{X} \to \mathcal{P}(\mathcal{A})$, which is a mapping from the observation space $\mathcal{O}$ and constraint specification $\mathcal{X}$ to the distributions over actions $\mathcal{A}$. Let $\gamma \in (0, 1)$ denote a discount factor, $\mu(\mathcal{S})$ denote the initial state distribution, and $\tau$ denote a trajectory sequence of observations and actions induced by a policy $\pi$, *i.e.,* $\tau = (o_0, a_0, o_1, \cdots)$. For any given $x$, we seek a policy $\pi$ that maximizes the cumulative discounted reward $J_R$ while keeping the cumulative discounted cost $J_C$ below a specified cost constraint threshold $h_C(x)$:

$$\max_{\pi} \quad J_R(\pi) \doteq \mathop{\mathbb{E}}_{\tau \sim \pi} \left[ \sum_{t=0}^{\infty} \gamma^t R(s_t, a_t) \right] \quad \text{s.t.} \quad J_C(\pi) \doteq \mathop{\mathbb{E}}_{\tau \sim \pi} \left[ \sum_{t=0}^{\infty} \gamma^t C(s_t, a_t, x) \right] \leq h_C(x),$$

where $\tau \sim \pi$ is shorthand for indicating that the distribution over trajectories depends on $\pi$ : $s_0 \sim \mu, o_t \sim Z(\cdot|s_t), a_t \sim \pi(\cdot|o_t, x), s_{t+1} \sim T(\cdot|s_t, a_t)$. We use $C(s_t, a_t, x)$ and $h_C(x)$ here to emphasize that both functions depend on the particular constraint specification $x$.

**Task setup.** Our goal is to show that constraints specified in natural language allow for generalization to new tasks that require similar constraints during learning. With this in mind, we consider the following safety training and safety evaluation setup:

**(1) Safety training:** During training, we generate random environment layouts and starting states $s_0$ while keeping the reward function $R$ fixed. For each episode, we randomly generate a constraint function $C$ and limit $h_C$. We then sample a constraint text $x$ that describes $C$ and $h_C$ from the training set of texts. The constraint text $x$ is an input to the agent's policy. Whenever the agent violates a constraint (at any step), it is provided with a scalar cost penalty learned by the model from $C(s, a, x)$. The agent, therefore, sees a variety of different task layouts and constraints, and learns a policy with respect to the constraints for this task as well as how to interpret textual constraints.

**(2) Safety evaluation:** During evaluation, we place the agent in new environments with randomly generated layouts, with a different reward function $R'$. The set of possible constraints $C$ is the same as seen in training, but the corresponding constraint texts are from an unseen test set. During this phase, the agent is not provided any cost penalties from the task. This setup allows us to measure two things: **(1)** how well an agent can learn new tasks while following previously learned textual constraints, and **(2)** the applicability of our method when using textual constraints unseen in training.

## 4 HAZARDWORLD

To our knowledge, there do not currently exist datasets for evaluating RL agents that obey textual constraints.[3] Thus, we design a new benchmark called HAZARDWORLD in which the agent starts

---

[3]Even though there are several instruction following tasks, our task setup is different, as mentioned previously.

each episode at a random location within a procedurally generated environment and receives a textual constraint $x$, sampled from a pool of available constraints. The agent's goal is to collect all the reward-providing entities while adhering to the specified constraint. Other than the constraint specified, the agent has complete freedom and is not told about how to reach reward-providing states.

HAZARDWORLD contains three types of constraints–**(1)** *budgetary constraints*, which impose a limit on the number of times a set of states can be visited, **(2)** *relational constraints*, which define a minimal distance that must be maintained between the agent and a set of entities, and **(3)** *sequential constraints*, which are constraints that activate unsafe states when a specific condition has been met. In total, we collect 984 textual constraints for HAZARDWORLD-grid (GridWorld environment) and 2,381 textual constraints for HAZARDWORLD-robot (robotic tasks). Table 1 provides examples.

**HAZARDWORLD-grid.** We implement HAZARDWORLD-grid (Table 1(a)) atop the 2D GridWorld layout of BabyAI [17, 56]. We randomly place three *reward entities* on the map: '*ball*,' '*box*,' and '*key*,' with rewards of 1, 2, and 3, respectively. We also randomly place several *cost entities* on the map: '*lava*,' '*water*,' and '*grass*'. We give a cost penalty of 1 when agents step onto any cost entities, which are specified using a textual constraint $x$. The entire state $s_t$ is a grid of size $13 \times 13$, including the walls, and the agent's observation $o_t$ is a $7 \times 7$ grid of its local surroundings. There are 4 actions–for moving up, down, left and right. We use the deterministic transition here.

*Train-test split.* We generate two disjoint training and evaluation datasets $\mathcal{D}_{\text{train}}$ and $\mathcal{D}_{\text{eval}}$. $\mathcal{D}_{\text{train}}$ consists of 10,000 randomly generated maps paired with 80% of the textual constraints (787 constraints overall), *i.e.,* on average each constraint is paired with 12.70 different maps. $\mathcal{D}_{\text{eval}}$ consists of 5,000 randomly generated maps paired with the remaining 20% of the textual constraints (197 constraints), *i.e.,* on average one constraint is paired with 25.38 maps. In $\mathcal{D}_{\text{eval}}$ we change the rewards for ball, box, and key to 1, 2, and -3, respectively. Therefore, in $\mathcal{D}_{\text{eval}}$, the agent has to avoid collecting the key to maximize reward.

**HAZARDWORLD-robot.** We build HAZARDWORLD-robot (Table. 1(b)) atop the SAFETY GYM environment [18] to show the applicability of our model to tasks involving high-dimensional continuous observations. In this environment, there are five constraint entities paired with textual constraints: *hazards* (dark blue puddles), *vases* (stationary but movable teal cubes), *pillars* (immovable cylinders), *buttons* (touchable orange spheres), and *gremlins* (moving purple cubes). This task is more challenging than the 2D case since some obstacles are constantly moving. The agent receives a reward of 4 for reaching a goal position and a cost penalty of 1 for bumping into any constraint entities. The observation $o_t$ is a vector of size 109, including coordinate location, velocity of the agent, and observations from lidar rays that detect the distance to entities. The agent has two actions–control signals applied to the actuators to make it move forward or rotate. The transitions are all deterministic.

*Train-test split.* We follow the same process for obtaining a train-test split as in HAZARDWORLD-grid. $\mathcal{D}_{\text{train}}$ consists of 10,000 randomly generated maps paired with 80% of textual constrains (1,905 constraints), *i.e.,* on average one constraint is paired with 5.25 maps. $\mathcal{D}_{\text{eval}}$ consists of 1,000 randomly generated maps paired with the remaining 20% of textual constrains (476 constraints), *i.e.,* on average one constraint is paired with 2.10 maps. In $\mathcal{D}_{\text{eval}}$ we add four additional goal locations to each map (*i.e.,* the maximum reward is 20). The agent has to learn to navigate to these new locations.

**Data collection.** For the textual constraints in both environments, we collected free-form text in English using Amazon Mechanical Turk (AMT) [57]. To generate a constraint for HAZARDWORLD, we provided workers with a description and picture of the environment, the cost entity to be avoided, and one of the following: **(a)** the cost budget (budgetary), **(b)** the minimum safe distance (relational), or **(c)** the other cost entity impacted by past events (sequential). We then cleaned the collected text by writing a keyword matching script followed by manual verification to ensure the constraints are valid.

## 5 Learning to Interpret Textual Constraints

We seek to train agents that can adhere to textual constraints even when learning policies for new tasks with different reward structures. We now describe our model and training and evaluation procedures.

### 5.1 Model

We design the RL agent as a deep neural network that consists of two parts (Fig. 2)–**(1)** a *constraint interpreter* which processes the text into structured safety criteria (a constraint mask and threshold)

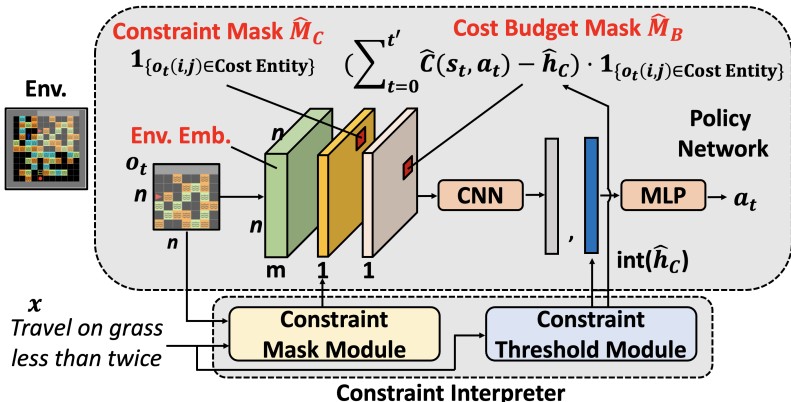

Figure 2: **Model overview.** Our model consists of two parts: **(1)** the *constraint interpreter* produces a constraint mask and cost constraint threshold prediction from a textual constraint and an observation, **(2)** a *policy network* takes in these presentations and produces a constraint-satisfying policy.

and **(2)** a *policy network* which uses the output of the interpreter along with observations to produce an action. For simplicity, in the following descriptions, we assume state $s$ and observation $o$ to be 2D matrices, although the model can easily be extended to other input representations.

**(1) Constraint interpreter (Fig. 3).** We concatenate an observation embedding of size $n \times n \times m$ from observations $o$ of size $n \times n$ with the embedding of the textual constraints $x$ of size $l$ from a long-short-term-memory (LSTM), followed by using a convolutional neural network (CNN) to get an embedding vector. We use this vector to produce a constraint mask $\hat{M}_C$, a binary matrix with the same dimension as $o$–each cell of the matrix is $0/1$ depending on whether the model believes the absence or presence of a constraint-related entity (*e.g., 'lava'*) in the corresponding cell of the observation $o$. In addition, we feed the textual constraints into an LSTM to produce $\hat{h}_C$, a real-valued scalar which predicts the constraint threshold, *i.e.,* the number of times an unsafe state is allowed.

For the case of sequential constraints with long-term dependency of the past events, $\hat{M}_C$ will depend on the past states visited by the agent. For example, in Fig. 3(b), after the agent visits '*water*', $\hat{M}_C$ starts to locate the cost entity ('*grass*'). Thus, for sequential constraints, we modify the interpreter by adding an LSTM layer before computing $\hat{M}_C$ to take the state history into account. Using $\hat{M}_C$ and $\hat{h}_C$ allows us to embed textual constraints in the policy network.

**(2) Policy network.** The policy network produces an action using the state observation $o_t$ and the safety criteria produced by the constraint interpreter. The environment embedding is concatenated with the constraint mask $\hat{M}_C$ (predicted by the constraint interpreter) and a *cost budget mask*, denoted by $\hat{M}_B$. The cost budget mask is derived from $\hat{h}_C$ (also predicted by the constraint interpreter) and keeps track of the number of constraint violations that the agent has made in the past over the threshold. $\hat{M}_B$ is an $n \times n$ matrix where each element takes the value of $\sum_{t=0}^{t'} \hat{C}(s_t, a_t; x) - \hat{h}_C$ (*i.e.,* the value of constraint violations past the budget until $t'$th step) if there is a cost entity in $o_t(i, j)$, or zero otherwise. During the safety evaluation phase, we estimate the cumulative cost $\sum_{t=0}^{t'} \hat{C}(s_t, a_t; x)$ using the predicted $\hat{M}_C$ and the agent's current location at time $t$. After concatenating both the constraint mask $\hat{M}_C$ and cost budget mask $\hat{M}_B$ to the observation embedding, we then feed the resulting tensor into CNN to obtain a vector (grey in Fig. 2). This vector is concatenated with a vectorized $\text{int}(\hat{h}_C)$ (*i.e., $\hat{h}_C$* rounded down) and fed into an MLP to produce an action.

**POLCO in HAZARDWORLD-robot.** To apply POLCO in this environment, the constraint interpreter predicts the cost entity given the textual constraints. We then map the cost entity to the pre-defined embedding vector (*i.e.,* one-hot encoding). We then concatenate the embedding vector, the embeddings of the predicted $\hat{h}_C$, and the value of cost budget (rounded down) to the observation vector. Finally, the policy network takes in this concatenated observation and produces a safe action.

**Advantages of the design.** The design of POLCO tightly incorporates textual constraints into the policy network. Our model factorization–into **(1)** a constraint interpreter and **(2)** a policy network–

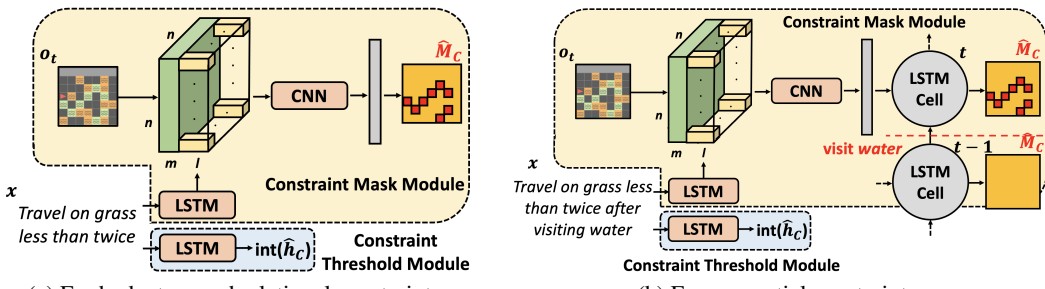

(a) For budgetary and relational constraints      (b) For sequential constraints

Figure 3: **Constraint interpreter.** **(a)** For the budgetary and relational constraints, a constraint mask module takes the environment embedding and text vector representation as inputs and predicts $\hat{M}_C$. **(b)** For the sequential constraints, we use an LSTM to store the information of the past visited states. For these three types of constraints, we use another LSTM given $x$ to predict $\hat{h}_C$.

allows us to design specific constraint interpreters for different types of constraints.[4] Furthermore, our approach scales gracefully to handling multiple constraints. While existing safe RL algorithms require retraining the policy for each unique combination of constraints, we can simply add together the $\hat{M}_C$ of each constraint to handle multiple constraints imposed simultaneously.

## 5.2 Safety training

We first train the constraint interpreter using a random policy to collect trajectories, and then we use the trained interpreter to predict constraints while training the policy network.

**Stage 1: Interpreter learning.** We use a random policy to explore the environment, and obtain trajectories consisting of observations $o_t$, along with the corresponding textual constraint $x$. Using the constraint violations encountered in the trajectory and the cost specification $C$, we obtain a target $M_C$ for training the constraint interpreter. In addition, we also obtain the ground-truth value of $\hat{h}_C$ for learning the constraint threshold module.

We train the constraint mask module of the constraint interpreter by minimizing the following binary cross-entropy loss over these trajectories: $\mathcal{L}(\Theta_1) = -\mathbb{E}_{(o_t,x)\sim\mathcal{D}_{\text{train}}}\Big[\frac{1}{|M_C|}\sum_{i,j=1}^n y\log\hat{y} + (1 - y)\log(1 - \hat{y})\Big]$, where $y$ is the target $M_C(i,j;o_t,x)$, which denotes the target (binary) mask label in $i$th row and $j$th column of the $n \times n$ observation $o_t$, $\hat{y}$ is the predicted $\hat{M}_C(i,j;o_t,x)$, *i.e.,* the probability prediction of constraint mask, and $\Theta_1$ are the parameters of the constraint mask module.

For the constraint threshold module, we minimize the following loss: $\mathcal{L}(\Theta_2) = \mathbb{E}_{(o_t,x)\sim\mathcal{D}_{\text{train}}}\big[(h_C(x) - \hat{h}_C(x))^2\big]$, where $\Theta_2$ are the parameters of the constraint threshold module.

This approach ensures cost satisfaction *during both policy learning and safety evaluation*, an important feature of safe RL. If we train both the policy and the interpreter simultaneously, then we risk optimizing according to inaccurate $\hat{M}_C$ and $\hat{h}_C$ values, as observed in our experiments.

**Stage 2: Policy learning.** We use a safe RL algorithm called projection-based constrained policy optimization (PCPO) [9] to train the policy network. During training, the agent interacts with the environment to obtain rewards and penalty costs ($\hat{M}_C$) are provided from the trained constraint interpreter for computing $J_R(\pi)$ and $J_C(\pi)$ (ground-truth $C$ is not used). PCPO is an iterative method that performs two key steps in each iteration[5]–optimize the policy according to reward and project the policy to a set of policies that satisfy the constraint. During safety evaluation, we evaluate our model in the new task with the new reward function and the textual constraints from $\mathcal{D}_{\text{eval}}$.

---

[4]$\hat{M}_B$ equates to a scaled up version of $\hat{M}_C$ since we assume only one constraint specification per episode, but this is not necessary in general since we may have multiple constraints over different cost entities. In that case, $\hat{M}_B$ may have different cost budgets for different cells (entities).

[5]One can use other safe RL algorithms such as Constrained Policy Optimization (CPO) [7]

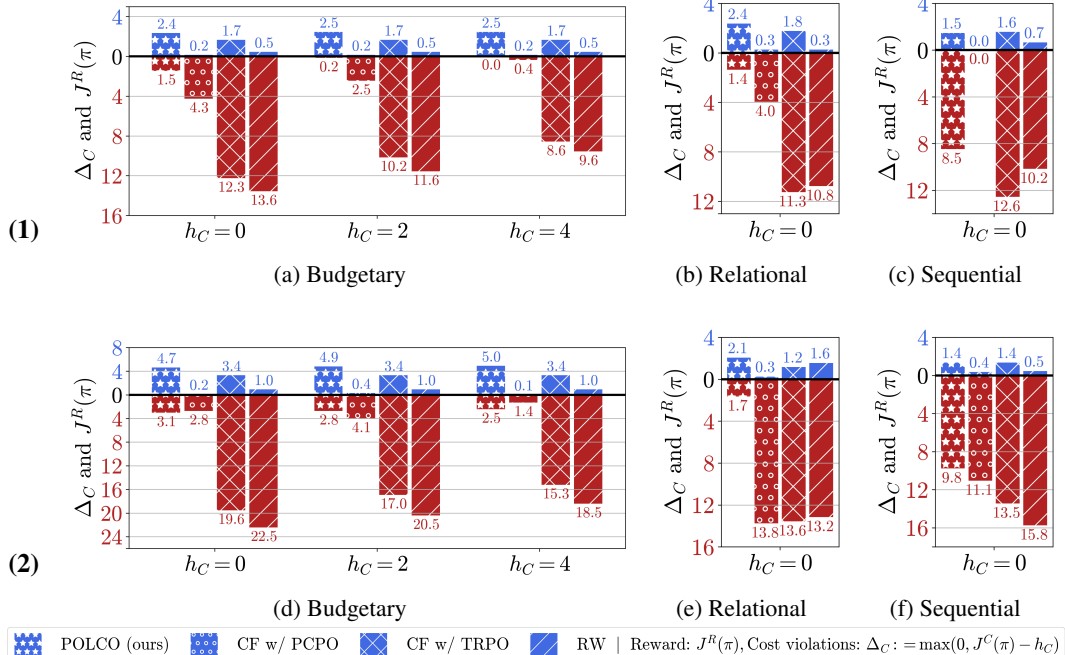

Figure 4: Results in HAZARDWORLD-grid over different values of $h_C$. These graphs represent the results of budgetary, relational, and sequential constraints, respectively. The **blue bars** are the reward performance ($J^R(\pi)$) and the **red bars** are the constraint violations ($\Delta_C$). For $J^R(\pi)$, higher values are better and for $\Delta_C$, lower values are better. **(1)** Results for transfer to the new tasks. **(2)** Results for handling multiple textual constraints. POLCO generalizes to unseen reward structure and handle multiple constraints with minimal constraint violations in the new task.

## 5.3 Safety evaluation

**(1) Transfer to new tasks:** We take the policy trained in $\mathcal{D}_{\text{train}}$ and fine-tune it on tasks having new reward functions with textual constraints from $\mathcal{D}_{\text{eval}}$. We *do not* retrain the constraint interpreter on $\mathcal{D}_{\text{eval}}$. The policy is fine-tuned to complete the new tasks without the penalty signals from the cost function $C$. In HAZARDWORLD-robot, we optimize the policy using CPO [7].

**(2) Handling multiple textual constraints:** We also test the ability of our model to handle multiple constraints imposed simultaneously (from $\mathcal{D}_{\text{eval}}$), by adding the cost constraint masks $\hat{M}_C$ of each constraint together when given multiple constraints. During safety training, the policy is still trained with a single constraint. No fine-tuning is performed and the reward function is maintained the same across training and evaluation in this case.

## 6 Experiments

Our experiments aim to study the following questions: **(1)** Does the policy network, using representations from the constraint interpreter, achieve fewer constraint violations in new tasks with different reward functions? **(2)** How does each component in POLCO affect its performance?

### 6.1 Setup

**Baselines.** We consider the following baselines:
**(1)** *Constraint-Fusion (CF) with PCPO*: This model [58] takes a concatenation of the observations and text representations as inputs (without $M_C$, $M_B$ and $h_C$) and produces an action, trained with an end-to-end approach using PCPO. This model jointly processes the observations and the constraints.
**(2)** *CF with TRPO:* We train CF using trust region policy optimization (TRPO) [59], which *ignores* all constraints and only optimizes the reward. This is to demonstrate that the agent will have substantial

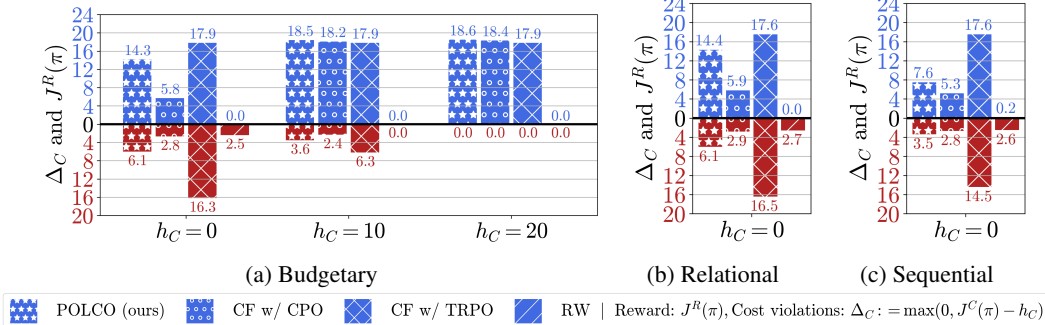

Figure 5: Results in HAZARDWORLD-robot over different values of $h_C$ for transfer to the new tasks. POLCO achieves competitive results with higher rewards and lower cost violations.

constraint violations when ignoring constraints.

**(3)** *Random Walk (RW)*: We also include a random walk (RW) baseline, where the agent samples actions uniformly at random.

**Evaluation metrics.** To evaluate models, we use **(1)** the average value of the reward $J_R(\pi)$, and **(2)** the average constraint violations $\Delta_C := \max(0, J_C(\pi) - h_C)$. Good models should have a small $\Delta_C$ (*i.e.,* close to zero) while maximizing $J_R(\pi)$. More details on the implementation, hyper-parameters, and computational resources are included in the Appendix A,B, and C.

### 6.2 Results

**HAZARDWORLD-grid.** Fig. 4(1) shows results for all models in the first evaluation setting of transfer to new tasks. POLCO has lower constraint violations in excess of $h_C$ while still achieving better reward performance in all cases. In comparison, the high cost values ($\Delta_C$) obtained by RW and CF with TRPO indicate the challenges of task. This supports our idea of using the learned constraint interpreter to learn a new task with similar textual constraints while ensuring constraint satisfaction. CF with PCPO has higher constraint violations, and in the most cases, does not optimize the reward, which suggests that it cannot transfer the constraint understanding learned in $\mathcal{D}_{\text{train}}$ to $\mathcal{D}_{\text{eval}}$.

Fig. 4(2) shows our evaluation with multiple textual constraints. We see that POLCO achieves superior reward and cost performance compared to the baselines, while CF with PCPO has worse reward and cost performance. This shows that our approach is flexible enough to impose multiple constraints than that of existing safe RL methods which requires retraining the policy for each unique combination of constraints.

**HAZARDWORLD-robot.** Fig. 5 shows transfer to new tasks in HAZARDWORLD-robot. The $J^R(\pi)$ and $\Delta_C$ of RW is relatively small since the agent does not move much because of random force applied to each actuator. For the budgetary constraints, although CF with TRPO achieves the best reward when $h_C = 0$, it has very large constraint violations. POLCO performs better than the baselines–it induces policies with higher reward under fewer constraint violations in most cases. In contrast, CF with CPO has lower reward performance.

Having demonstrated the overall effectiveness of POLCO, our remaining experiments analyze **(1)** the learned models' performance evaluated on the same reward function as in $\mathcal{D}_{\text{train}}$, and **(2)** the importance of each component–$M_B, M_C$ and $h_C$ embedding in POLCO. For compactness, we restrict our consideration in HAZARDWORLD-grid.

**Evaluation with reward function from $\mathcal{D}_{\text{train}}$.** To provide another point of comparison in addition to our main results, we evaluate all models using the same reward function as in $\mathcal{D}_{\text{train}}$, but with unseen textual constraints from $\mathcal{D}_{\text{eval}}$. ( Fig. 6) We observe POLCO achieves the lowest violations across different choices of $h_C$ compared to the baselines. This implies that merely combining the observations and the text is not sufficient to learn an effective representation for parsing the constraints. In addition, POLCO achieves the best reward performance under cost satisfaction for the more complex relational and sequential constraints. For the relational case, although the CF agent trained with PCPO satisfies the constraints, it has a relatively low reward.

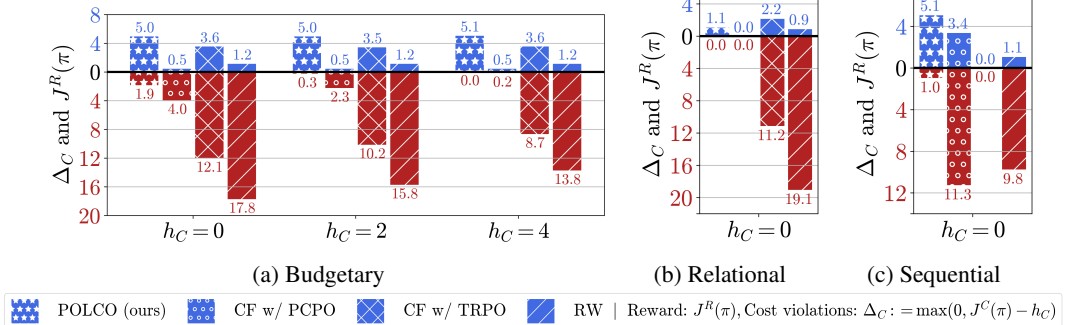

(a) Budgetary      (b) Relational    (c) Sequential

POLCO (ours)    CF w/ PCPO    CF w/ TRPO    RW | Reward: $J^R(\pi)$, Cost violations: $\Delta_C := \max(0, J^C(\pi) - h_C)$

Figure 6: Results in HAZARDWORLD-grid for the setting of evaluation with the same reward function as seen in training. POLCO achieves higher reward and lower constraint violations over the baselines.

**Ablation studies.** We also examine the importance of each part in POLCO (Fig. 7). To eliminate prediction errors from the constraint interpreter, we use the *true* $M_C$ and $h_C$ here. Our full model achieves the best performance in all cases, averaging 5.12% more reward and 2.22% fewer constraint violations. Without $M_C$, the agent cannot recognize cost entities effectively, which causes the agent to incur 66.67% higher $\Delta_C$ compared with the full model (which has a $\Delta_C$ close to zero). This shows that $h_C$ embedding and the $M_B$ mask are useful in enabling constraint satisfaction given textual constraints.

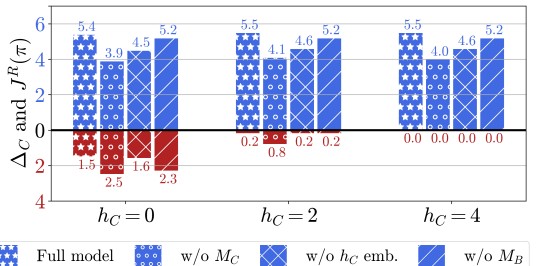

Figure 7: Ablations showing the effect of each component in POLCO for the budgetary constraint.

## 7 Conclusion

Our work provides a view towards machines that can interoperate with humans. As autonomous agents proliferate into our world, they should be able to understand safety constraints set by human agents around them. Accordingly, we proposed the problem of safe RL with natural language constraints, created a new benchmark called HAZARDWORLD to test agents and developed a new algorithm for the task (POLCO) that learns to interpret constraints. Our paper defines and trains machine agents that understand what *not to do* in natural language, much like instruction following tasks enable agents in understanding what *to do*.

The thesis of our POLCO approach is that modularity enables reuse. By bootstrapping a modular constraint interpreter through exploration, our model scales easily to multiple constraints and to shifts in the environment's reward structure, all while exploring new environments safely. We applied POLCO within HAZARDWORLD to train an agent that navigates safely by obeying natural language constraints. This agent is a step towards creating applications like cleaning robots that can obey free form constraints, such as "*don't get too close to the TV*" – a relational constraint in our formulation.

No model is without limitations. The absolute scores of POLCO on HAZARDWORLD still leave a lot of room for improvement using better models or training techniques. The current version of HAZARDWORLD is also not all-encompassing – we envision it as a benchmark that evolves over time, with the addition of new types of constraints and new environments. Future work can investigate training without explicit labels for the constraint interpreter, potentially using techniques like Gumbel softmax [60], or extending POLCO to tasks with more realistic visuals.

**Acknowledgments**

The authors would like to thank members of the Princeton NLP Group, the anonymous reviewers, and the area chair for their comments and feedback. Tsung-Yen Yang thanks Siemens Corporation, Corporate Technology for their support.

**Funding Transparency Statement**

Funding in direct support of this work: scholarship by Siemens Corporation, Corporate Technology, and the Princeton CSML Summer Research Award.

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
