# Supplementary Material

**Outline.** The supplementary material is outlined as follows. Section A details the dataset and the procedure of collecting the dataset. Section B describes the parameters of the constraint interpreter and the policy network, and the PCPO training details. Section C provides the learning curves of training the policy network. Section D details how we apply POLCO in the robotics tasks. Finally, dataset and code to reproduce our experiments are available at `https://sites.google.com/view/polco-hazard-world/`.

## A  Dataset

At a high level, HAZARDWORLD applies the instruction following paradigm to safe reinforcement learning. Concretely, this means that safety constraints in our environment are specified via language. Our dataset is thus comprised of two components: the environment, made up of the objects that the agent interacts with, and the constraint, which imposes a restriction on which environmental states can be visited.

The environment is procedurally generated. For each episode, HAZARDWORLD places the agent at a randomized start location and fills the environment with objects. HAZARDWORLD then randomly samples one constraint out of all possible constraints and assigns this constraint to the environment.

We collected natural language constraints in a two-step process. In the first step, or the data generation step, we prompted workers on Amazon Mechanical Turk with scenarios shown in Fig. 8. Workers are provided the minimum necessary information to define the constraint and asked to describe the situation to another person. For example, to generate a so-called budgetary constraint, workers are given the cost entity to avoid (*'lava', 'grass'* or *'water'*) and the budget (*i.e.,* $h_C$, a number 0 through 5). The workers use this information to write an instruction for another person. This allows us to ensure that the texts we collected are free-form. These generations form our language constraints.

In the second step, or the data validation step, we employed an undergraduate student to remove invalid constraints. We define a constraint as invalid if (a) the constraint is off-topic or (b) the constraint does not clearly describe states that should be avoided. Examples of valid and invalid constraints are included in Table 2. Finally, we randomly split the dataset into 80% training and 20% test sets. In total, we spent about $ 1500 for constructing HAZARDWORLD.

In HAZARDWORLD and Lawawall, the agent has 4 actions in total: $a \in \mathcal{A} = \{\text{right}, \text{left}, \text{up}, \text{down}\}$. The transition dynamics $T$ is deterministic.

## B  Architectures, Parameters, and Training Details

**Policy network in POLCO.** The architecture of the policy network is shown in Fig. 9. The environment embedding for the observation $o_t$ is of the size $7{\times}7{\times}3$. This embedding is further concatenated with the cost constraint mask $M_C$ and the cost budget mask $M_B$. This forms the input with the size $7{\times}7{\times}5$. We then use convolutions, followed by dense layers to get a vector with the size 5. This vector is further concatenated with the $h_C$ embedding. Finally, we use dense layers to the categorical distribution with four classes (*i.e.,* turn right, left, up or down in HAZARDWORLD). We then sample an action from this distribution.

**Constraint interpreter in POLCO.** The architecture of the constraint interpreter is shown in Fig. 10. For the constraint mask module, the input is the text with $w$ words. We then use an embedding network, followed by an LSTM to obtain the text embedding with the size 5. The text embedding is duplicated to get a tensor with the size $7{\times}7{\times}5$. This tensor is concatenated with the observation of size $7 \times 7 \times 3$, creating a tensor with the size $7{\times}7{\times}8$. In addition, we use a convolution, followed by dense layers and a reshaping to get the cost constraint mask $M_C$.

Next, we use a heuristic to compute $\hat{C}_{tot} := \sum_{t=0}^{t'} C(s_t, a_t; x)$ from $M_C$. At execution time, we give our constraint interpreter access to the agent's actions. We initialize $\hat{C}_{tot} = 0$. Per timestep, our agent either turns or moves forward. If the agent moves forward and the square in front of the agent contains a cost entity according to $M_C$, we increment $\hat{C}_{tot}$.

Here, your friend is navigating the environment below to collect the ball, box, and key. The environment also contains lava, water, and grass blocks.

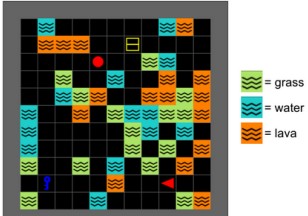

≋ = grass

≋ = water

≋ = lava

(a) General prompt for all constraint classes.

One of these block types causes wear and tear, so your friend can only stand on them a limited number of times. For example, if **Number of Times = 3**, then your friend can only walk on the **Dangerous Block** three times, at most. Knowing this, tell the other person what she can or cannot do in this situation.

**Dangerous Block:** lava

**Number of Times:** 5

Type what you would say here...

(b) Budgetary prompt.

One of these objects is dangerous, and your friend must maintain a minimum safe distance from that type of object. For example, if **Minimum Distance = 1**, then your friend must stay at least 1 step away from **Dangerous Blocks** at all times. Knowing this, tell the other person what she can or cannot do in this situation.

**Dangerous Block:** lava

**Minimum Distance:** 3

Type what you would say here...

(c) Relational prompt.

One of the three block types is a trigger block. If you friend touches a trigger block, one of the block types becomes dangerous. Your job is to tell the other person what she can or cannot do in this situation. For example, if **Trigger Block = lava**, then your friend can't walk on **Dangerous Blocks** after walking on any **lava** blocks.

**Trigger Block:** lava

**Dangerous Block:** water

Type what you would say here...

(d) Sequential prompt.

Figure 8: AMT workers receive the general prompt and one of the three specific prompts. They are then asked to instruct another person for the given situation. This ensures that the texts we collected are free-form.

For the constraint threshold module, we use the same architecture to get the text embedding. We then use dense layers to predict the value of $h_C$.

**Details of the algorithm–PCPO.** We use a KL divergence projection in PCPO to project the policy onto the cost constraint set since it has a better performance than $L_2$ norm projection. We use GAE-$\lambda$ approach [61] to estimate $A_R^\pi(s, a)$ and $A_C^\pi(s, a)$. We use neural network baselines with the same architecture and activation functions as the policy networks. The hyperparameters of training POLCO are in Table 3. We conduct the experiments on the machine with Intel Core i7-4770HQ CPU. The experiments are implemented in rllab [62], a tool for developing RL algorithms.

**Baseline model–Constraint Fusion (CF).** The model is illustrated in Fig. 11. An LSTM takes the text $x$ as an input and produces a vector representation. The CNN takes the environment embedding of $o_t$ as an input and produces a vector representation. These two vector representations are concatenated, followed by a MLP to produce an action $a_t$. We do not consider other baselines in [43] and [5]. This is because that their models are designed to learn a multi-modal representation (*e.g.,* processing a 3D vision) and follow goal instructions. In contrast, our work focuses on learning a constraint-satisfying policy.

The parameters of the baseline is shown in Fig. 12. We use the same CNN parameters as in our policy network to process $o_t$. Then, we use the same LSTM parameters as in our constraint mask module to get a vector representation with size 5. Note that we use almost the same number of the

| Constraint Type | Examples |
|---|---|
| Budgetary | The water should only be stepped on a max of 5 times. |
| | Lava hurts a lot, but you have special shoes that you can use to walk on it, but only up to 5 times, remember! |
| | You can get in lava, but only once. |
| | Four is the most number of times you can touch water |
| | You cannot step on the lava block at all. You will die otherwise. |
| Relational | Water will hurt you if you are two steps or less from them. |
| | Always stay 1 step away from lava |
| | Any block within one unit of a grass cannot be touched. |
| | The explosion radius of grass is three, so stay at least that distance away from grass. |
| | Waters are dangerous, so do not cross them. |
| Sequential | Make sure you don't walk on water after walking on grass. |
| | Do not touch the water or water will become risky. |
| | You may touch the water first, but the lava is dangerous so do not touch it after. |
| | Avoid lava since you can only walk on it once. After that the lava will hurt you. |
| | Water will trigger grass to become dangerous. |
| Invalid | good |
| | move foreward |
| | Just avoid the perimeter when collecting the objects, and you'll be safe. |
| | Your directions are as follows: if you're facing a block with a water block in front of it, walk five blocks ahead . . . (81 more words) |
| | asdf |

Table 2: Examples from the various constraint classes. When a constraint does not fully describe all forbidden states in the environment, we classify it as invalid.

| Parameter | |
|---|---|
| Reward dis. factor $\gamma$ | 0.99 |
| Constraint cost dis. factor $\gamma_C$ | 1.0 |
| step size $\delta$ | $10^{-3}$ |
| $\lambda_R^{\text{GAE}}$ | 0.95 |
| $\lambda_C^{\text{GAE}}$ | 0.9 |
| Batch size | 10,000 |
| Rollout length | 200 |
| Number of policy updates | 2,500 |

Table 3: Parameters used in POLCO.

parameters to ensure that POLCO does not have an advantage over CF. Finally, we use dense layers to the categorical distribution with four classes. We then sample an action from this distribution.

## C   Additional Experiments

**Learning curves of training the policy network.**   The learning curves of the undiscounted constraint cost, the discounted reward, and the number of steps over policy updates are shown for all tested algorithms and the constrains in Fig. 13. Overall, we observe that

(1) POLCO improves the reward performance while satisfying the cost constraints during training in all cases,

(2) the policy network trained with TRPO has substantial cost constraint violations during training,

(3) the policy network trained with FPO is overly restricted, hindering the reward improvement.

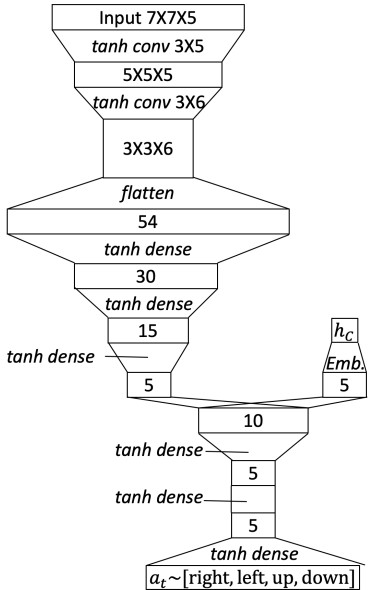

Figure 9: Description of the policy network in POLCO.

# D  POLCO for 3D robotics tasks.

To deal with pixel observations $o_t$, we can still use the proposed architecture to process $o_t$ as shown in Fig. 14. To predict the cost constraint mask $\hat{M}_C$, we use the object segmentation method to get the bounding box of each object in the scene. As a result, the area of that bounding box will be one if there is a cost entity (*i.e.,* the forbidden states mentioned in the text). Otherwise, the bounding box contains a zero. For $\hat{M}_B$, we can use a similar approach to compute the cumulative cost violations at each step. In addition, to deal with navigation environments with 3D ego-centric observations, we propose shifting the $o_t$, $\hat{M}_C$ and $\hat{M}_B$ matrices to be the first-person view. The bounding box approach for image case can still be applied here. We leave this proposal to future work.


Figure 10: Description of the constraint interpreter.

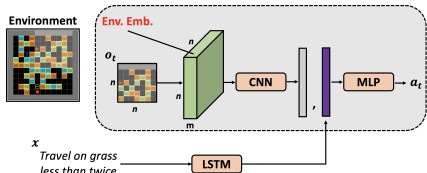

Figure 11: Baseline model–Constraint Fusion (CF). It is composed of two parts – **(1)** a CNN takes $o_t$ as an input and produce a vector representation, **(2)** an LSTM takes $x$ as an input and produce a vector representation. We then concatenate these two vectors, followed by a MLP to produce an action $a_t$.

(b) Did you specify all the training details (e.g., data splits, hyperparameters, how they were chosen)? [Yes] See Section 6 and the supplementary material.

(c) Did you report error bars (e.g., with respect to the random seed after running experiments multiple times)? [N/A] We follow the same style of machine learning papers to report the results.

(d) Did you include the total amount of compute and the type of resources used (e.g., type of GPUs, internal cluster, or cloud provider)? [Yes] See the supplementary material.

4. If you are using existing assets (e.g., code, data, models) or curating/releasing new assets...

(a) If your work uses existing assets, did you cite the creators? [Yes] See the supplementary material.

(b) Did you mention the license of the assets? [Yes] They are open-sourced.

(c) Did you include any new assets either in the supplemental material or as a URL? [Yes] See the supplementary material.

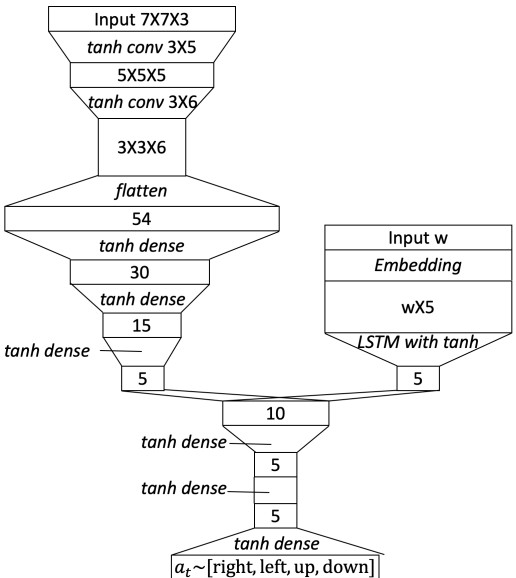

Figure 12: Description of our baseline model-Constraint Fusion (CF).

(d) Did you discuss whether and how consent was obtained from people whose data you're using/curating? [Yes]  We obtained consent to use worker-generated data via Amazon Mechanical Turk.

(e) Did you discuss whether the data you are using/curating contains personally identifiable information or offensive content?  [Yes]  We check the data and do not find any identifiable information or offensive content.

5. If you used crowdsourcing or conducted research with human subjects...

(a) Did you include the full text of instructions given to participants and screenshots, if applicable? [Yes]  See the supplementary material.

(b) Did you describe any potential participant risks, with links to Institutional Review Board (IRB) approvals, if applicable? [N/A]

(c) Did you include the estimated hourly wage paid to participants and the total amount spent on participant compensation? [Yes]  See the supplementary material.

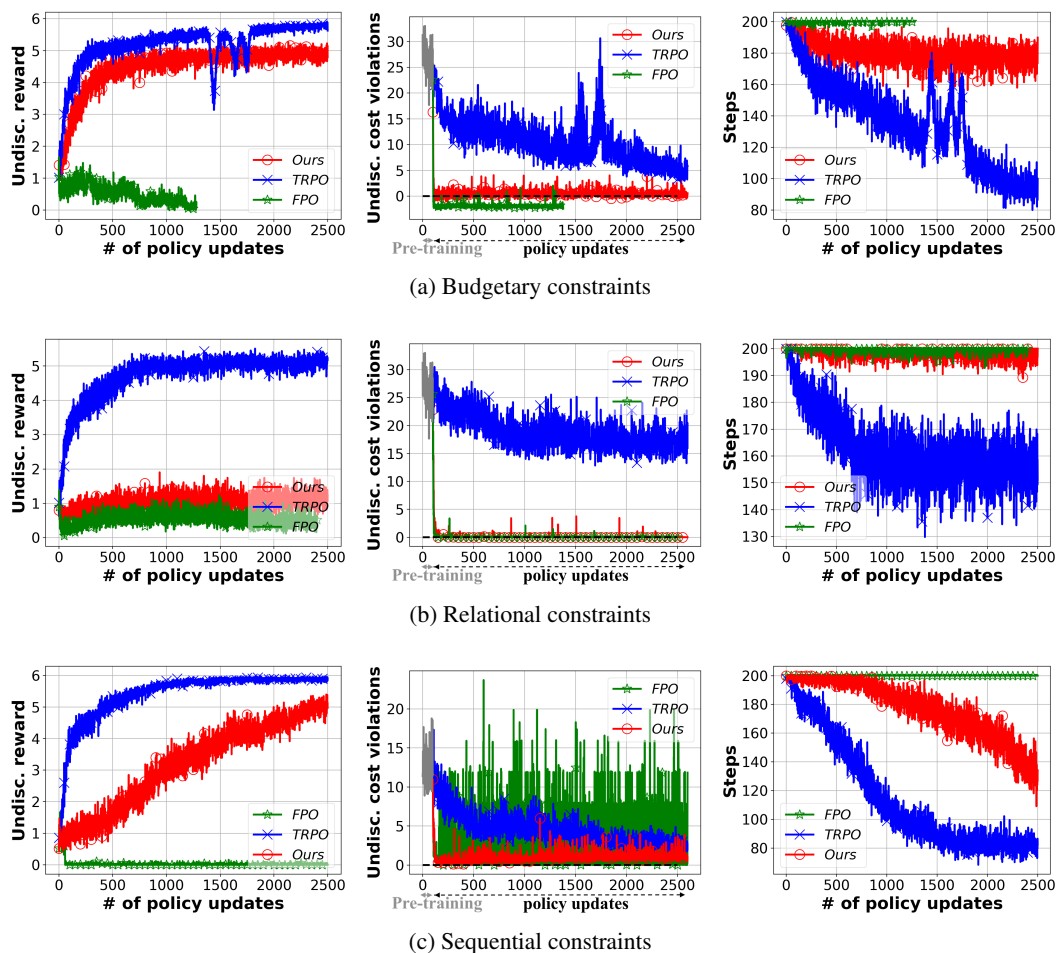

(a) Budgetary constraints

(b) Relational constraints

(c) Sequential constraints

Figure 13: **Learning curves of training the policy network.** The undiscounted reward, the undiscounted cost violations (*i.e.,* $\Delta_C = J_C(\pi) - h_C$), and the number of steps over policy updates for the tested algorithms and the constrains. In the undiscounted cost violations plots, we further include the numbers for the interpreter pre-training stage in the first 100 points. This is equal to 5000 trajectories. The maximum allowable step for each trajectory is 200. We observe that POLCO satisfies the cost constraints throughout training while improving the reward. In contrast, the policy network trained with TRPO suffers from violating the constraints and the one trained with FPO cannot effectively improve the reward. (Best viewed in color.)

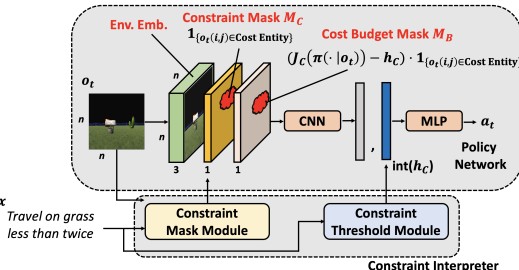

Figure 14: POLCO for pixel observations and 3D ego-centric observations. The red cloud area represents the bounding box of each object in $o_t$.