# OpenReview forum: "Safe Reinforcement Learning with Natural Language Constraints"
_NeurIPS.cc/2021/Conference — NeurIPS 2021 Spotlight_

### Official Review · Reviewer_8Sns · 2021-07-08

**Rating:** 8
**Confidence:** 4

**Summary:**

This paper contributes to the area of constrained RL by adding machinery to interpret and apply constraints specified in natural language.  They also design and implement a testbed world to evaluate such systems.  In this world, they demonstrate that their approach consistently outperforms other methods, both in cumulative reward and constraint violations.  They also use an ablation study to demonstrate the need for each component of their approach.


**Ethical Concerns:**

I don't see any ethical concerns in this work.


**Limitations And Societal Impact:**

The authors briefly describe limitations of their work on page 9.

They state in the Checklist that they do not see potential negative impacts, but do not address it in the main body of the manuscript.


**Main Review:**

This paper is a clear contribution to the field and merits publication.

One thing that I would like to see, however, is a comparison with other constrained RL approaches that are given the constraints in their native formats, e.g., logic rules, finite automata, etc.  It's reasonable that this paper's approach might not always win in these tests, but they still maintain the advantage of NL-formatted constraints.


**Time Spent Reviewing:**

1

---

> ### Author Response · Authors · 2021-08-10
> **Response**
>
> We thank the reviewer for the helpful and insightful feedback. We provide answers to individual questions below.
>
> 1. “One thing that I would like to see, however, is a comparison with other constrained RL approaches that are given the constraints in their native formats, e.g., logic rules, finite automata, etc. It's reasonable that this paper's approach might not always win in these tests, but they still maintain the advantage of NL-formatted constraints.”
>
> Ans: Thanks for pointing this out. In our ablation studies in Fig. 7, we evaluate the full model using the logic rules (i.e., ground truth $M_C$ and $h_C$) in the test set with the same reward function as in the training set. In addition, we evaluate the full model with the same setting but using language constraints in Fig. 6. We can see that in the budgetary constraints, the model with the logic rules achieves 8% higher rewards and 21% fewer cost violations than the one with language constraints given $h_C=0$. Same trend can be found in $h_C=2$ and $h_C=4$.
>
> Furthermore, we also ran additional experiments on the same setup for the relational and sequential constraints. We observe a similar tendency: the one with logic rules has 10% higher rewards and 25% fewer cost violations on average. This indeed shows the point that the proposed approach does not outperform the one with the logic rules (can be seen as upper bounds on performance), but still preserve the easily accessible medium for describing the constraints, which is the main point of this paper. We will add the discussion in the revised version.
>
> 2. “They state in the Checklist that they do not see potential negative impacts, but do not address it in the main body of the manuscript.”
>
> Ans: Thank you for pointing this out. We will refine our presentation in the revised version.

---

### Official Review · Reviewer_YDss · 2021-07-15

**Rating:** 7
**Confidence:** 4

**Summary:**

This paper presents a new multi-task environment for learning the agent with natural language constraints. The author breaks away from existing approaches that specify the constraints in mathematical form and extends it to specify the constraints in natural language. Their environment provides grid-world and robotics environments, and various kinds of constraints for safe RL. Finally, the authors propose a model for constraint policy optimization applicable to language constraint settings and show the generalization effect of language constraints.

**Limitations And Societal Impact:**

Yes, the authors have adequately addressed it.

**Main Review:**

This paper is well written and addresses an important problem of language constraints for safe RL. I agree that language constraints are intuitive and easy-accessible, and I think specifying the constraints in natural language can provide various advantages when applying the constraint to a real-world problem. The detailed comments and questions are as follows:

1. In the result of experiments (Figure 4, 5, 6, 7) what is the maximum performance that can be achieved while satisfying the cost constraint? It would be helpful to understand how well the proposed algorithm performs.

2. The authors define three types of constraint and are modeling two types to consider different types of constraint. Is there any reason to model this way? (Is it not possible with a single model to consider all constraints as the same?)

3. In safety training, the authors mention that PCPO was used for policy learning (page 7 lines 260-261), and in safety evaluation, CPO was used for HazardWorld-robot (page 7 line 271). Is there any reason to use different methods?

4. (Page 9, lines 326-337) In the ablation studies, it is necessary to add a detailed explanation, such as exactly which domain was tested.


**Time Spent Reviewing:**

10

---

> ### Author Response · Authors · 2021-08-10
> **Response**
>
> We thank the reviewer for the helpful and insightful feedback. We provide answers to individual questions below.
>
> 1. “In the result of experiments (Figure 4, 5, 6, 7) what is the maximum performance that can be achieved while satisfying the cost constraint? It would be helpful to understand how well the proposed algorithm performs.”
>
> Ans: The maximum rewards with zero cost violations are: 3 for Fig. 4(a), and 6 for Fig. 4(b), Fig. 6, and Fig. 7. For HazardWorld-robot (Fig. 5), the agent receives the reward of 4 when reaching a goal location at each step. Depending on the distance between the start and goal locations and the duration in the goal position, the average maximum reward is 30.
>
> 2. “The authors define three types of constraint and are modeling two types to consider different types of constraint. Is there any reason to model this way? (Is it not possible with a single model to consider all constraints as the same?)”
>
> Ans: The reason to have separate models for handling non-sequential and sequential constraints is to streamline the analysis of the proposed approach. In addition, we agree that it is possible to have a single constraint interpreter for all constraints. For example, we can use transformers to handle non-sequential and sequential constraints. We will add discussions in the revised version.
>
> 3. “In safety training, the authors mention that PCPO was used for policy learning (page 7 lines 260-261), and in safety evaluation, CPO was used for HazardWorld-robot (page 7 line 271). Is there any reason to use different methods?”
>
> Ans: One can use any safe reinforcement learning algorithms to train the model. The reason for using CPO for HazardWorld-robot is that CPO appears to have smoother reward improvement than PCPO in HazardWorld-robot during learning. We conjecture that for the high dimensional observation and large action space (as in HazardWorld-robot), we need to have a small step size to update the policy, and CPO tends to produce a smaller step size than PCPO due to the projection step in PCPO, which causes a large update size. We will mention this in the revised version.
>
> 4. “In the ablation studies, it is necessary to add a detailed explanation, such as exactly which domain was tested.”
>
> Ans: Thank you for pointing this out. The ablation studies in Fig. 7 were tested in HazardWorld-grid with the same reward function during training and evaluation. We will add the details in the revised version.

---

> > ### Comment · Reviewer_YDss · 2021-08-24
> > **Response to rebuttal**
> >
> > Thank you for taking the time to respond to my questions. I've read other reviews and the author's rebuttal. I still think that this paper is well-motivated and has a clear contribution. I maintain my original score.

---

### Official Review · Reviewer_v8rs · 2021-07-15

**Rating:** 6
**Confidence:** 4

**Summary:**

The paper is proposing an end-to-end reinforced approach of sequential task learning incorporating constraint using the natural language called POLCO standing for Policy Optimization with Language Constraints.

The authors propose a novel grid-world and robotic manipulation environment, called HazardWorld, to illustrate their proposal. Each world is composed of a variety of cells symbolizing a different level of safe reachability, in the case of robotic manipulation the state is discretized for applying the proposedapproach. Then, a set of three different constraint types expressed in natural language are studied: Budgetary, Relational, Sequential and correspond to different types and levels of compositionality.

Constraints are translated into reachability space through a CNN encoder and transfer into the policy and can be learnt in an end-to-end manner.  Reaching an unsafe place would produce a large negative reward. Proposing this setting, natural language is seamlessly incorporated into a non-reachable space of the considered environment.

The paper introduces a two-step process for safety policy training. First, a random policy explores the environment and grasp reward in accordance to language specified set of constraints. A constraint mask over the environment definition is computed using binary cross-entropy. Second, the policy is trained using PCPO which allow optimizing a policy by maximizing the reward function while satisfying the learnt constraint mask through re-projection in the satisfiability policy subspace.

The overall policy architecture is an original composition of state of the art differentiable building blocks suitable for differentiable training.

**Ethical Concerns:**

No Ethical issue to mention.

**Limitations And Societal Impact:**

No section is explicitly addressing these concerns. However, as the question of the safety of sequential decision algorithms is the subject of this paper, it seems largely related to this subject of concern.

**Main Review:**

The paper addresses an important problem of sequential decision learning which is safety at training and test time and its seamless specification.
The proposed method proposes to use an interesting modality which is the natural language for this matter.
The model is trained in a differentiable manner and is tested on a large set of possible constraints expressing various levels of compositional in language.
The two-step learning process is novel and intuitive and the policy architecture is leveraging the know components of differentiable programming.
The experiments show convincing results of the capability of the method to handle the various types of constraint, however at the cost of an initial random exploration, which may be considered as contradictory to the overall purpose of the work.
The main concern is about the generalization of such an approach including stochastic and/or partially observable environments.
At the current stage, the paper hardly allows appreciating the possible impact of the approach, especially the constraint mask module, on such a more realistic scenario.


**Time Spent Reviewing:**

2.5hours

---

> ### Author Response · Authors · 2021-08-10
> **Response**
>
> We thank the reviewer for the helpful and insightful feedback. We provide answers to individual questions below.
>
> 1. “...at the cost of an initial random exploration, which may be considered as contradictory to the overall purpose of the work.”
>
> Ans: We agree that at the beginning of the training, the agent must visit the forbidden states in order to conduct language grounding. Without any prior knowledge (e.g., the system dynamics, cost function), it is impossible to ensure safety for randomly initialized RL agents during training. However, we emphasize that once we train the constraint interpreter, we can reuse it for training a new task (different reward function/collecting new entities) with minimal cost violations. By obtaining world knowledge through random exploration, we can then perform safe reinforcement learning. We describe this setup in line 49 and line 142.
>
> 2. “The main concern is about the generalization of such an approach including stochastic and/or partially observable environments.”
>
> Ans: Our policy architecture can handle stochastic environments since the cost interpreter and the constraint mask does not depend on the transition dynamics of the environment. Our discrete grid environments are partially observable–the agent observes in a square in front of it, but not behind. To handle partial observability in more complex environments, we can use object segmentation techniques to identify unsafe states (do not get close to the hot stove or slippery floor) in the scene, which is analogous to the constraint mask. We discuss the extension to partially observable environments in appendix line 689, and leave it for future work.
>
> 3. “At the current stage, the paper hardly allows appreciating the possible impact of the approach, especially the constraint mask module, on such a more realistic scenario.”
>
> Ans: We attempt to demonstrate the applicability of our approach by creating and testing our agents on both discrete (grid world) and continuous (mujoco) environments for this novel task setup. As we discussed in the previous response, we can use object segmentation methods to extract unsafe states from a first-person view. This allows us to generalize our approach in more realistic settings such as navigation in the 3D world (Chen et al., 2020). Further, our main goal is to describe this new task setup and provide an appropriate benchmark for the community to develop more advanced algorithms and methods in the future.
> Chen et al., TOUCHDOWN: Natural Language Navigation and Spatial Reasoning in Visual Street Environments, 2020.
>
> 4. “No section is explicitly addressing these concerns. However, as the question of the safety of sequential decision algorithms is the subject of this paper, it seems largely related to this subject of concern.”
>
> Ans:  We summarize the concerns of the proposed model here. The absolute scores on HarzardWorld still leave a lot of room for improvement. Future work can investigate training POLCO without explicit labels for the constraint interpreter, potentially using techniques like Gumbel softmax, or extending POLCO to tasks with more realistic visuals.
>
> The societal impact of this work is a view towards robots and machine learning agents that understand natural language. With this work, we hope to enable agents that understand what $\textbf{not to do}$ in natural language, much like instruction following enables agents that understand what $\textbf{to do}$. We have discussed the limitation of the work in the conclusion (line 350). We will refine our presentation and highlight concerns in the revised version.

---

### Author Response · Authors · 2021-08-10
**General response**

Thank you so much for the constructive comments. We will refine the presentation and add in details as suggested.

---

### Decision · Program_Chairs · 2021-09-27

**Decision:**

Accept (Spotlight)

**Comment:**

This paper presents a framework wherein constraints (particularly safety constraints) on a reinforcement learning agent can be specified using natural language. The paper is well written, clear, and was well-received by the reviewers, all of whom recommend acceptance. The AC finds the idea of natural language constraints particularly compelling, and strongly supports acceptance.

One additional related work is "Preventing undesirable of intelligent machines", which proposes an "interface" that the users of ML/RL algorithms can use to define safety constraints. It seems like the method proposed here would plug-in directly as such an "interface", making this paper a valuable contribution to Seldonian methods (perhaps beyond RL) as well.